# The Effect of Mealworm Frass on the Chemical and Microbiological Properties of Horticultural Peat in an Incubation Experiment

**DOI:** 10.3390/ijerph20010021

**Published:** 2022-12-20

**Authors:** Anna Nogalska, Sebastian Wojciech Przemieniecki, Sławomir Józef Krzebietke, Dariusz Załuski, Agnieszka Kosewska, Małgorzata Skwierawska, Stanisław Sienkiewicz

**Affiliations:** 1Department of Agricultural and Environmental Chemistry, Faculty of Agriculture and Forestry, University of Warmia and Mazury in Olsztyn, Oczapowskiego 8, 10-719 Olsztyn, Poland; 2Department of Entomology, Phytopathology and Molecular Diagnostics, Faculty of Agriculture and Forestry, University of Warmia and Mazury in Olsztyn, Prawocheńskiego 17, 10-719 Olsztyn, Poland; 3Department of Genetics, Plant Breeding and Bioresource Engineering, Faculty of Agriculture and Forestry, University of Warmia and Mazury in Olsztyn, Plac Łódzki 3, 10-719 Olsztyn, Poland

**Keywords:** *Tenebrio molitor* L., frass, urea (46% N), deacidified peat, nutrients, pH, EC

## Abstract

Insect farming is growing in popularity, and in addition to insect meal, it generates waste products such as exuviae and frass, which can be recycled in agriculture. The aim of this incubation experiment was to evaluate the effect of *Tenebrio molitor* L. frass on selected chemical and biological properties of deacidified peat, which is widely used in horticulture. The optimal rate of frass fertilizer in peat for growing vegetables and ornamental plants was determined, with special emphasis on mineral nitrogen levels. Peat was fertilized with five nitrogen rates, 0, 50, 100, 200, and 400 mg dm^−3^, and supplied with frass or urea. The study demonstrated that frass can be used as organic fertilizer. An increase in the nitrogen rate significantly increased mineral nitrogen content and electrical conductivity and decreased Ca content in peat. Both frass and urea increased the ammonification rate at the beginning of incubation and the nitrification rate from the second week of the experiment. Higher frass rates (5 and 10 g dm^−3^) increased the content of plant-available nutrients (nitrogen, phosphorus, magnesium, potassium, and sodium) in peat as well as the abundance of microorganisms supporting organic matter mineralization. Unlike frass, urea increased the counts of nitrogen-fixing bacteria in peat.

## 1. Introduction

The rapid growth of the world population, which is expected to exceed 9 billion in 2050 according to the United Nations Food and Agriculture Organization (FAO), will massively increase the demand for food and will further contribute to the depletion of the Earth’s resources [1]. It is estimated that by 2050, the global demand for meat will increase by 75% relative to 2005 [2]. The use of edible insects as food was proposed as a potential solution to this problem during a FAO meeting in 2012 [3]. Insects are widely consumed by several hundred million people around the world, mainly in Latin America, Africa, and Asia [4], and they are highly abundant in protein, fat, and minerals [5]. The popularity of insect protein is also on the rise in Europe due to growing meat production costs, the associated greenhouse gas emissions, as well as ethical and health concerns. In countries with a warm climate, insects are larger and easier to acquire, but in Europe, edible insects have to be produced in dedicated farms [6,7,8]. The interest in insect farming increased considerably after edible insects had been approved for human and animal consumption in the European Union [9,10]. Unlike homeothermic livestock, insects are poikilotherms, and they do not rely on metabolic processes to maintain their body temperature within narrow boundaries [11]. As a result, insect farming requires considerably less energy and water per gram of harvested protein in comparison with livestock production. Insect farms also occupy less space and are more environmentally friendly due to lower greenhouse gas and ammonia emissions [12,13]. Insects grow much faster than poultry, sheep, and pigs, and they convert feed into body mass up to 12 times more efficiently than cattle. The mealworm (*Tenebrio molitor* L.) is the most widely farmed species of edible insects. The growing popularity of mealworms can be attributed to their high nutritional value and a high content of available protein [14,15,16,17,18]. Mealworm meal can be also used as fertilizer, with a high content of organic nitrogen. Przemieniecki et al. [19] demonstrated that mealworm meal fertilizer positively affected the growth of wheat seedlings and the abundance of beneficial soil and rhizosphere microorganisms. In addition to meal, insect farming also generates waste products such as exuviae and frass that can be recycled in agriculture. Houben et al. [20,21] found that frass can partly or completely replace mineral NPK fertilizers because it is rapidly mineralized to release plant-available nutrients. According to research, frass improves soil fertility and promotes plant growth [22,23,24,25]. Kagata and Ohgushi [23] reported that frass increased the content of plant-available nitrogen in soil. The fertilizing properties of frass are also influenced by insect diets. Frass rich in N-NH_4_ was obtained from insects fed nitrogen-rich plants [26]. The content of mineral nitrogen can decrease in the initial stage of frass decomposition because it is immobilized by soil-dwelling microorganisms or evacuated from soil with volatized ammonia [27]. Soil is a complex matrix whose physical, chemical, and biological properties are affected by soil-dwelling microorganisms [28]. Soil microbes play important roles by stabilizing biological processes in the soil solution, converting nutrients to plant-available forms, producing humus, promoting plant growth, and protecting plants against pests and phytopathogens. Functional microorganisms are effective indicators of soil health. Frass is highly abundant in nitrogen, carbon, and other nutrients [20,23], and it stimulates soil-dwelling microorganisms responsible for ammonification, nitrification, denitrification, and nitrogen fixation. At the microbiological level, these processes determine the forms, losses, and availability of organic nitrogen supplied to the soil. Changes in the carbon-to-nitrogen ratio directly affect total saprotroph counts and fungal-to-bacterial ratios. Therefore, the applied fertilizer should stimulate the growth of beneficial bacteria, including antagonistic bacteria, and decrease the abundance of phytopathogenic fungi. According to Berggren et al. [29], the fertilizing potential of frass should be thoroughly examined before it is applied on an industrial scale. Zero waste solutions are urgently needed to mitigate the rapid depletion of natural resources, rising costs of conventional mineral fertilizers produced mainly from fossil fuels, and instability in countries with the largest energy reserves. Improper or excessive use of mineral fertilizers decreases soil fertility by stimulating acidification, disrupting the ionic balance, suppressing microbial activity, introducing harmful substances to soil, and inducing salinity. At present, the main goal of fertilization should not be to maximize yields but to ensure the high quality of crops and to maintain soil fertility with a minimum impact on the natural environment.

The chemical composition of insect diets (mainly vegetables and cereal flakes) should be analyzed to ensure that frass is safe for the environment. Insect frass is a rich source of nitrogen, macronutrients, and micronutrients; it is also dry, friable, and odorless. Therefore, it can be used as organic fertilizer, in particular in the production of vegetables and ornamental plants whose quality, nutritional value, or esthetic appeal are important for consumers. The use of mealworm frass as fertilizer in horticulture and hobby gardening remains insufficiently investigated. Therefore, the aim of this incubation experiment was to evaluate the effect of *Tenebrio molitor* L. frass on selected chemical and biological properties of deacidified peat, which is easily available and widely used in horticulture and in the production of vegetables and ornamental plants in home gardens. The content of nutrients, in particular nitrogen, was compared in frass and urea (46% N) as the conventional source of nitrogen. The optimal rate of frass fertilizer in a horticultural substrate for growing vegetables and ornamental plants was determined, with special emphasis on mineral nitrogen levels.

## 2. Materials and Methods

### 2.1. Characterization of Frass and Horticultural Peat

Loose mealworm (*Tenebrio molitor* L.) frass was obtained from the Tenebria mealworm farm (Poland). Mealworms were fed a friable diet composed of wheat flakes, wheat bran, yeast, and rice flour (in proportions that constitute a trade secret), as well as fresh carrots administered at a 1:1 ratio to the friable feed mix. Frass and urea were combined with deacidified peat and incubated. The chemical composition of peat and frass is presented in Appendix A.

### 2.2. Experimental Design

Deacidified peat with the addition of mealworm (*Tenebrio molitor* L.) frass or urea was incubated for nine weeks during a laboratory experiment conducted in 2022 at the Department of Agricultural and Environmental Chemistry of the University of Warmia and Mazury in Olsztyn, Poland. Peat was fertilized with five nitrogen rates of 0, 50, 100, 200, and 400 mg dm^−3^, supplied with frass or urea, in three replicates. Loose frass in the amount of 1.32, 2.63, 5.26, and 10.52 g dm^−3^ and dissolved urea (10 cm^3^) at a concentration of 0.11, 0.22, 0.44, and 0.88 g dm^−3^ were thoroughly combined with 1 dm^3^ of the horticultural substrate. Distilled water (10 cm^3^) was added to unfertilized treatments and treatments fertilized with frass. Peat was incubated for nine weeks in closed and air-tight PET containers at a stable temperature of 26 °C and substrate moisture content equivalent to 60% of the maximum water-holding capacity. During incubation, peat samples were collected for chemical and microbiological analyses. The first sample was collected at the end of the first week of incubation, and successive samples were collected at two-week intervals (five sampling sessions in total). Samples of horticultural peat mixed with frass or urea were also collected for microbiological analyses on the first day of the experiment.

### 2.3. Chemical Analysis of Frass and Horticultural Substrate

Frass samples were dried to absolute dry weight at 105 °C, weighed, and wet mineralized in concentrated sulfuric acid (VI), with hydrogen peroxide (H_2_O_2_) as the oxidizing agent (Speed Digester K-439; BÜCHI, Labortechnik AG, Flavil, Switzerland). The N content of mineralized frass was determined by the Kjeldahl method (KjelFlex K-360; BÜCHI Labortechnik AG, Flavil, Switzerland); phosphorus content was determined by the vanadium-molybdenum method (UV-1201 V spectrophotometer, Shimadzu Corporation, Kyoto, Japan); magnesium content was determined by atomic absorption spectrometry (AAS) (AAS1, Carl Zeiss, Jena, Germany); and calcium and potassium content was determined by atomic emission spectroscopy (AES) (Phlavo 4, Carl Zeiss, Jena, Germany).

Chemical analyses of deacidified peat (before the experiment) and incubated peat were performed after extraction with 0.03 M dm^−3^ CH_3_COOH (universal extraction method described by Nowosielski [30]) at a 1:10 ratio. In the obtained extracts, N-NH_4_ content was determined with Nessler’s reagent; N-NO_3_ content was determined with phenoldisulfonic acid (UV-1201 V spectrophotometer, Shimadzu Corporation, Kyoto, Japan); P content was determined by the vanadium–molybdenum method (UV-1201 V spectrophotometer, Shimadzu Corporation, Kyoto, Japan); K, Ca, and Na content was determined by atomic emission spectroscopy (AES) (BWB Technologies UK Ltd., Hambridge Ln, Newbury, The United Kingdom); Mg content was determined by atomic absorption spectrometry (AAS) (AAS1, Carl Zeiss, Jena, Germany); and S-SO_4_ and Cl^−^ content was determined by the nephelometric method (UV-1201 V spectrophotometer, Shimadzu Corporation, Kyoto, Japan).

The pH and electrical conductivity (EC) of peat were determined by potentiometric titration in distilled water (soil:H_2_O ratio of 1:2.5) (CP-505 pH meter, Elmetron Sp. j. Zabrze, Poland). The content of C_org_ in peat and frass was determined with the Vario Max Cube CN elemental analyzer (Elementar Analysensysteme GmbH, Langenselbold, Germany).

### 2.4. Quantitative PCR of Microbial Functional Genes

DNA was isolated with the Soil DNA Purification Kit (EURx, Gdańsk, Poland). Soil samples were ground in a mortar. After pre-grinding, 100 mg of soil was transferred to a 2 mL tube containing glass beads and a lysis buffer, and was homogenized in a TissueLyser LT bead mill (Qiagen, Hilden, Germany). Cells were lysed for 10 min at maximum speed. Successive steps of the DNA isolation procedure were based on the instructions attached to the Soil DNA Purification Kit. Quantitative PCR setup is presented in Appendix A [31,32,33,34,35,36,37,38,39,40,41] and in previous studies [19,42].

The qPCR standards were prepared based on the appropriate amplicons of *Bacillus subtilis*, *Pseudomonas putida*, and *Fusarium culmorum.* The environmental material was ligated with plasmids (TOPO™ TA Cloning™ Kit, with pCR™ 2.1-TOPO™, Thermo Fisher Scientific, Waltham, MA, USA) and used as the standard for the respective domains. All reactions were performed in samples with a volume of 20 µL each, using the Probe or SybrGreen qPCR mastermix 2× (A&A Biotechnology, Gdynia, Poland). Reaction efficiency was determined at 0.94–1.03 (R^2^ = 0.996–1).

### 2.5. Statistical Analysis

The results were processed statistically in Statistica 13.3 (TIBCO, Palo Alto, CA, USA) at a significance level of *α* = 0.05 [43]. In the first stage, the results were subjected to Friedman’s repeated measures analysis of variance by rank with Iman-Davenport modification [44]. The significance of differences between frass and urea fertilizers was determined by the Mann–Whitney test. The significance of differences between five nitrogen rates (0, 50, 100, 200, and 400 mg dm^−3^) in each fertilizer was determined by Dunn’s test with Bonferroni correction. In the second stage, all examined chemical and microbiological properties were assessed by principal component analysis (PCA) with the use of a heatmap and a clustering tree.

## 3. Results

### 3.1. Chemical Changes in Deacidified Peat

It was found that mealworm frass and urea applied at five rates (0, 50, 100, 200, and 400 mg N dm^−3^) significantly differentiated the chemical properties of deacidified peat. Regardless of the applied fertilizer, the content of ammonium nitrogen (N-NH_4_) and nitrate nitrogen (N-NO_3_) in horticultural peat increased significantly with a rise in the nitrogen rate (Table 1). The highest increase (more than three-fold) in the content of mineral nitrogen (N-NH_4_ + N-NO_3_) was observed after the application of 50 mg N dm^−3^ relative to unfertilized peat, and each of the higher nitrogen rates nearly doubled the content of mineral nitrogen in the substrate, relative to the previous rate.

The average content of N-NH_4_ and N-NO_3_ in peat incubated with frass was 2.5- and 1.5-fold lower, respectively, than in peat incubated with urea. In peat incubated with frass and urea, the content of mineral nitrogen increased significantly with a rise in the nitrogen rate, excluding peat fertilized with frass at the lowest nitrogen rate (50 mg dm^−3^), where N-NH_4_ content was equivalent to that of unfertilized peat. Urea applied at 50 and 100 mg N dm^−3^ significantly increased the content of N-NH_4_ relative to treatments where equivalent amounts of nitrogen were supplied with frass.

The rate of changes in the content of mineral nitrogen in horticultural peat incubated with frass or urea for nine weeks is presented in Figure 1, Figure 2 and Figure 3. Irrespective of the applied fertilizer, the rate of nitrogen ammonification was high in the first days of incubation (Figure 1), and the rate of nitrification increased rapidly beginning from the second week of incubation (Figure 2).

Similarly to urea, frass was ammonified in the first two weeks of incubation. In peat fertilized with frass, the content of mineral nitrogen nearly doubled (from 40 to 77 mg dm^−3^) in the first two weeks, decreased to around 47 mg dm^−3^ in weeks 3–6, and increased to more than 60 mg dm^−3^ in weeks 7–9 (Figure 3).

Peat incubated with urea was characterized by smaller variation in the content of mineral nitrogen than peat incubated with frass, despite the fact that the content of plant-available N (in particular nitrate nitrogen) was much higher after the application of urea. In peat incubated with urea, mineral nitrogen levels were high already in the first week of the experiment (approx. 80 mg dm^−3^) and continued to increase up to week 5 to reach nearly 100 mg dm^−3^. The analyzed parameter decreased in week 7, and a repeated increase to 105 mg dm^−3^ was observed in the last two weeks of incubation (Figure 3).

Deacidified peat and frass were characterized by acidic pH before the experiment (pH in H_2_O = 5.7 and 5.6, respectively; Appendix A). Regardless of the applied fertilizer, all nitrogen rates significantly decreased the pH of the substrate relative to unfertilized peat (Table 2). It should also be noted that urea had a more acidifying effect on the tested substrate than frass, and the greatest decrease in peat pH (5.46) was observed in response to the highest urea rate.

The values of EC increased significantly with a rise in nitrogen rates. This parameter was determined at 0.11 mS cm^−1^ in unfertilized peat and 1.01 mS cm^−1^ in peat supplied with the highest nitrogen rate (Table 2). Electrical conductivity was significantly higher in peat fertilized with urea, in particular at 100 and 200 mg N dm^−3^. The observed increase resulted from a higher concentration of mineral nitrogen (N-NO_3_ and N-NH_4_) in peat after the application of urea than after application with frass. Ion concentrations in the horticultural substrate generally increased with a rise in the nitrogen rate, excluding Ca^2+^, whose concentration decreased. Peat fertilized with frass, especially at higher rates, was more abundant in plant-available phosphorus, magnesium, potassium, and sodium than peat fertilized with urea.

### 3.2. Microbiological Changes in Deacidified Peat

The study demonstrated that frass and urea exerted different effects on the abundance of total bacteria, total fungi, *Bacillus* spp., *Pseudomonas* spp., *chiA*, and *ureC* in peat (Table 3). Bacterial counts were estimated at 8.71 in peat fertilized with frass at 50–400 mg N dm^−3^ and at 8.61 in unfertilized peat. The abundance of bacteria decreased in response to urea applied at 200 mg N dm^−3^ compared with unfertilized peat. Total bacterial counts were significantly higher in peat fertilized with frass at 200 and 400 mg N dm^−3^ than in peat fertilized with urea.

Total fungal counts were lowest in the unfertilized treatment, moderate in peat fertilized with frass at 50–200 mg N dm^−3^, and highest in peat fertilized with frass at 400 mg N dm^−3^. In turn, the highest urea rate significantly decreased fungal counts relative to lower nitrogen rates. A comparison of the effects exerted by frass and urea on fungal counts in peat revealed significant differences when nitrogen was applied at 50, 200, and 400 mg dm^−3^. Frass stimulated the abundance of *Clostridium* spp., but a significant increase in the counts of these bacteria were noted only in peat supplied with frass at 100 and 400 mg N dm^−3^. In peat fertilized with urea at 200 mg N kg^−1^, no significant differences in *Clostridium* spp. counts were noted relative to the unfertilized treatment. The counts of *Bacillus* spp. increased significantly in response to rising frass rates, excluding the lowest rate (50 mg N dm^−3^). The highest urea rates (200 and 400 N dm^−3^) induced a significant decrease in *Bacillus* spp. counts in peat (by one order of magnitude), and the effects exerted by frass and urea were significantly different. The abundance of *Pseudomonas* spp. was significantly stimulated by frass (applied at 50–200 N dm^−3^), but a significant decrease in this parameter was observed in response to the highest nitrogen rate in comparison with unfertilized peat. The two highest urea rates decreased the abundance of *Pseudomonas* spp. All frass rates significantly increased *Pseudomonas* spp. counts relative to urea treatments. The number of *chiA* gene copies increased with a rise in the frass rate, and the abundance of *chiA* genes increased significantly beginning from the rate of 100 mg N dm^−3^. Urea decreased the counts of chitinolytic bacteria already at the lowest rate, but beginning from 100 mg N dm^−3^, urea induced a greater decrease in the abundance of *chiA* relative to the unfertilized treatment. The number of *nifH* gene copies increased in response to all frass rates. In turn, urea rates of 50 and 200 mg N dm^−3^ suppressed the growth of nitrogen-fixing bacteria but exerted a stimulatory effect when applied at 100 and 400 mg N dm^−3^. Neither the type nor the rate of fertilizer significantly affected the number of *nosZ* gene copies. The number of *amoA* gene copies increased with a rise in the nitrogen rate, regardless of fertilizer type. In peat incubated with frass, the number of *ureC* gene copies was similar to that noted in the unfertilized treatment, but the frass rate of 400 mg N dm^−3^ induced a significant decrease in this parameter in comparison with the remaining rates. The highest urea rates inhibited the growth of ureolytic bacteria. The number of *ureC* gene copies was significantly higher in peat fertilized with frass than urea at 100 and 200 mg N dm^−3^.

### 3.3. Relationships between the Chemical and Microbiological Parameters of Peat

Variation in the parameters of peat samples collected five times during the nine-week incubation experiment was examined by the PCA (Figure 4). The results of the PCA revealed that the content of N-NH_4_, N-NO_3_, and mineral nitrogen and the abundance of *amoA* (group I) were correlated and linked with increasing urea rates. In turn, the content of Cl^−^, S-SO_4_, Na, P, K, and Mg and the abundance of *Bacillus* spp., fungi, and *chiA* (group II) were correlated with the highest frass rates (200 and 400 mg N dm^−3^). High values of pH, Ca, *ureC,* and *Pseudomonas* spp. (group III) were noted in peat supplied with low rates of both fertilizers and in the unfertilized treatment. The abundance of *Clostridium* spp. described by the short vector was correlated with high nitrogen rates supplied with both fertilizers, whereas the abundance of *nosZ* was correlated with frass rates of 100 and 200 mg N dm^−3^. A negative correlation was observed between group I and group II results.

The studied parameters were visualized in a dendrogram and a heat map, which revealed that the nitrogen rate of 400 mg dm^−3^ exerted significantly different effects than the remaining rates (Figure 5). Nitrogen rates of 0–200 mg dm^−3^ produced three moderately different clades. The first clade was composed of unfertilized treatments, and it was characterized by high values of Ca and pH and a low content of Cl.

The second clade contained treatments fertilized with urea at 50–200 mg N dm^−3^, which were characterized by low or moderate values of the examined parameters. However, treatments supplied with urea at 50 and 100 mg N dm^−3^ were characterized by higher abundance of Clostridium spp., and peat fertilized with urea at 100 mg N dm^−3^ was characterized by higher abundance of *nifH*. In turn, the urea rate of 200 mg N dm^−3^ decreased the values of microbiological parameters, in particular the abundance of *ureC* and total bacteria.

The third clade comprised treatments fertilized with frass at 50–200 mg N dm^−3^. This group was characterized by high Ca content and high abundance of *nosZ*, *Pseudomonas* spp., total bacteria, total fungi, and *ureC*. However, the frass rate of 200 mg N dm^−3^ increased the values of S-SO_4_, Cl^−^, P, Mg, K, Na, EC, and *Bacillus* spp. relative to the remaining two rates in this clade. The fourth clade not only differed considerably from the remaining groups, but it was also characterized by substantial intra-clade variation, as demonstrated in the heatmap. The largest number of parameters with the highest values was observed in peat fertilized with frass at 400 mg N dm^−3^. This clade was characterized by high values of Cl^−^, N-NO_3_, N_min_, EC, *amoA,* and Clostridium spp. in treatments supplied with the highest rates of both fertilizers. Peat fertilized with the highest frass rate was characterized by high values of S-SO_4_, P, Mg, K, Na, *Bacillus* spp., *chiA*, total bacteria, and total fungi, whereas peat fertilized with urea at 400 mg N dm^−3^ was more abundant in N-NH_4_ and *nifH*.

## 4. Discussion

### 4.1. Chemical Changes in Deacidified Peat

Urea is rapidly hydrolyzed (ureolysis) by urease, a hydrolase class enzyme produced by microorganisms, usually within 2–14 days. Urea nitrogen becomes available to plants already after 2–3 days at optimal humidity (50–60%), neutral pH, and a temperature of 20–30 °C, but only after 20 days at suboptimal conditions (temperature below 8 °C) [45]. In turn, nitrogen from organic materials such as mealworm frass is mineralized to ammonium nitrogen during proteolysis (protein hydrolysis to peptides and amino acids) and ammonification (conversion of amino acids to ammonia). In the present study, a significant increase in mineral nitrogen levels was noted in peat incubated with frass, pointing to rapid mineralization. In turn, a significant decrease in N-NH_4_ content and an increase in N-NO_3_ content already after two weeks of incubation are indicative of rapid nitrification. Similar rates of mineralization and nitrification were reported by Houben et al. [20,21], who also noted a positive correlation between nitrogen mineralization and microbial activity. The cited authors also found that nitrification was the dominant process [20]. In turn, Kagata and Ohgushi [26] observed a considerable decrease in the content of ammonium nitrogen already in the first week of soil incubation with frass, and they attributed this loss to ammonia volatilization and nitrogen immobilization by soil microorganisms. In the current study, the concentration of mineral nitrogen, including nitrate nitrogen, decreased considerably in the seventh week of incubation, in particular in comparison with peat incubated with frass. In the work of Kowalska [46], a significant decrease in plant-available nitrate nitrogen on the 50th day of incubation was attributed to excessive growth of cellulolytic bacteria, which inhibited the development of nitrifying bacteria.

For field-grown vegetables, the optimal content of mineral nitrogen (N-NH_4_ + N-NO_3_) in soil is 50–130 mg dm^−3^ soil [45]. In the present incubation experiment, these requirements were met when peat was fertilized with the two highest frass rates (200 and 400 mg N dm^−3^) and with urea at 100 and 200 mg N dm^−3^. However, the recommended level of mineral nitrogen was exceeded (1.8-fold) when peat was fertilized with the highest urea rate. The recommended levels of nitrate nitrogen are much higher (120–300 mg dm^−3^ substrate) for vegetables and ornamental plants grown on organic and mineral substrates under cover [45]. In this case, only the highest frass and urea rates can supply the required amounts of nitrate nitrogen, whereas lower rates are insufficient and should be supplemented through nitrogen fertilization. However, nitrates tend to accumulate in vegetables, and fertilization rates in vegetable production should be controlled to minimize potential adverse health effects for consumers.

In the current study, all urea rates led to the acidification of the horticultural substrate. In contrast, low frass rates did not induce changes in peat pH, and pH decreased from 5.54 to 5.52 only in response to the two highest frass rates. Frass had a less acidifying effect than urea, probably because it contains alkaline elements (Ca and Mg). Houben et al. [20] attributed the decrease in soil pH to the acidic nature of frass and its rapid decomposition, which led to the release of CO_2_ and organic acids. However, they examined frass with a higher pH (5.8) than that analyzed in the present study (pH 5.6). The pH of substrates for growing most vegetables and ornamental plants should range from 5.5 to 6.5 [45]. Frass promoted the optimal peat pH, whereas the highest urea rate led to the acidification of the horticultural substrate.

In addition to the pH of horticultural substrates, which determines the availability of nutrients for plants, EC is also an important parameter that provides information about the concentrations of dissolved nutrients in the soil solution and soil salinity. Intensive mineral fertilization increases soil salinity due to excessive concentrations of NO_3_^−^, K^+^, Na^+^, Cl^−^, and SO_4_^2−^. High substrate salinity is often noted when plants are grown under cover. Excessive accumulation of K^+^ and SO_4_^2−^ and, to a lesser degree, Cl^−^ and Mg^2+^, followed by Ca^2+^, NH_4_^+^, and phosphate ions, is responsible for the salinization of horticultural substrates. Plant sensitivity to high soil salinity is determined by species, growth stage, soil moisture content, and production technology. The EC of organic and mineral substrates should not exceed 1.60 (bell peppers) and 1.90 mS cm^−1^ (cucumbers, tomatoes) for vegetables with a long growing season grown under cover, and 0.6 mS cm^−1^ for field-grown vegetables [45]. In the present study, EC was determined at 0.11 to 1.01 mS cm^−1^, and it was within the optimal range for most horticultural crops. Urea induced a significantly greater increase in the EC of deacidified peat than frass, which could be attributed to higher concentrations of mineral nitrogen (N-NO_3_ and N-NH_4_) in peat fertilized with urea. The above implies that mineral nitrogen was released more rapidly from urea than from frass. The content of the remaining nutrients (K, Na, Cl, S-SO_4_, Mg, Ca, and P) in peat was lower after the application of urea than frass. The concentrations of these nutrients generally increased with a rise in the nitrogen rate. Calcium was the only exception, which can be attributed to the fact that this element is bound by phosphate anions and adsorbed to form insoluble compounds.

### 4.2. The Effect of Frass and Urea on Functional Microorganisms in the Horticultural Substrate

Mealworm frass improved the microbiological quality of the horticultural substrate compared with urea. All tested frass rates increased total bacterial counts. Lower frass rates (50–100 mg N dm^−3^) stimulated the growth of *Pseudomonas* spp. and ureolytic bacteria, whereas higher rates (200 and 400 mg N dm^−3^) increased total fungal counts and the abundance of chitinolytic bacteria and *Bacillus* spp. Despite the above, clear correlations were not observed between the nitrogen rate and the abundance of *Clostridium* spp. and nitrogen-fixing bacteria, probably due to varied oxygen regimes in peat. Regardless of the applied fertilizer, the number of *amoA* gene copies increased with a rise in the nitrogen rate, which indicates that both fertilizers intensified nitrification processes. The tested fertilizers did not affect the abundance of denitrifying bacteria (*nosZ*).

This is one of the first studies to analyze the effect mealworm frass on the number of bacterial functional genes in a horticultural substrate, which is why the present results could not be compared with published findings. According to Poveda [47], the insect gut is naturally colonized by beneficial bacteria, which are transmitted to the soil environment with frass. Houben et al. [20] reported that soil fertilization with mealworm frass, alone or in combination with mineral NPK fertilizer, stimulated the activity of soil-dwelling microorganisms in comparison with mineral fertilizer alone. The greatest improvement in soil microbiological properties was noted after the application of frass with NPK fertilizer, namely rich sources of organic carbon (frass) and mineral nitrogen (NPK fertilizer). Similarly to the present study, frass increased the activity of nitrifying bacteria in the work of Houben et al. [20].

In peat fertilized with frass rates higher than 100 mg N dm^−3^, a significant and correlated increase was observed in the load of fungi, chitinolytic bacteria, and *Bacillus* spp., which suggests that fungi and antagonistic bacteria competed for limited resources in this ecological niche.

### 4.3. The Relationship between the Chemical and Microbiological Parameters of Peat Fertilized with Different Nitrogen Rates

The PCA revealed clear correlations between selected chemical and microbiological parameters of the horticultural substrate. Urea and frass exerted varied effects on deacidified peat, and frass strongly affected peat microbiota and mineralization processes. Frass and urea had a similar influence on N-NH_4_ levels. In peat fertilized with frass and urea, the abundance of the *amoA* gene was also correlated with N-NO_3_ content and high EC values. Frass rates higher than 100 mg N dm^−3^ caused a significant and correlated increase in P and K concentrations, which were partly correlated with microbiological parameters.

The results of the PCA were largely confirmed by the heatmap. Both frass and urea formed separate groups, and the highest rates of both fertilizers exerted the strongest, but varied, effects on the horticultural substrate. The nitrogen rate of 200 mg dm^−3^, in particular when supplied with frass, increased ion concentrations in the soil solution, EC values, and the abundance of potentially beneficial *Bacillus* spp. and ureolytic bacteria. The nitrogen rate of 400 mg dm^−3^ strongly influenced chemical parameters and radically changed the structure of soil microbial communities by increasing the counts of beneficial bacteria, including chitinolytic microorganisms. The abundance of bacteria converting ammonium nitrogen to nitrate nitrogen also increased, but the number of *nosZ* gene copies was generally low, which indicates that nitrates were not effectively converted to elemental nitrogen and that nitrogen losses were high. It should also be noted that the counts of nitrogen-fixing bacteria in peat increased in response to the highest urea rate. In the substrate fertilized with the highest frass rate, high N-NO_3_ content was strongly correlated with the number of *amoA* gene copies, whereas the content of mineral nitrogen was correlated with high abundance of fungi and *Bacillus* spp. (Figure 5).

Houben et al. [20] reported an improvement in the microbiological properties of soil after the application of frass + NPK fertilizer, which was attributed to the supply of organic matter and an improvement in the C:N:P ratio in the soil environment. The above treatment intensified nitrification processes, as demonstrated by rapid frass mineralization, increased release of CO_2_, and a decrease in soil pH. Similar observations were made in a study by Przemieniecki et al. [19], where the addition of mealworm (*T. molitor*) meal to soil increased the load of *amoA, chiA*, *Bacillus* spp., and fungi, which is consistent with the present findings. An ongoing greenhouse experiment (data not published) revealed that excessive fungal growth can compromise plant development at the beginning of the growing season by increasing the risk of seedling damping-off. However, the rapid growth of chitinolytic and antagonistic bacteria not only suppresses the adverse effects of soil fungi but also promotes the development of beneficial *Bacillus* spp. [48]. The present results expand the existing knowledge about the associations between functional bacterial groups and nutrients. Further research is needed to characterize the rhizosphere microbiome and determine its impact on the growth and development of plants fertilized with frass.

## 5. Conclusions

The present study demonstrated that frass is a potentially valuable and safe source of nutrients, in particular nitrogen, for plants. Deacidified peat fertilized with frass had a higher pH than peat fertilized with urea. The recommended content of mineral nitrogen in a horticultural substrate for the production of vegetables and ornamental plants was not exceeded in treatments supplied with high frass rates (200 and 400 mg N dm^−3^), which indicates that organic nitrogen was mineralized at a moderate rate and that nitrogen losses were minimized. However, due to periodic increases in soil salinity, the nitrogen rate of 400 mg dm^−3^ is not recommended for plants that are grown from seeds. In turn, the frass rate of around 10 g dm^−3^ substrate (400 mg N dm^−3^) supplied the optimal amounts of bioavailable N, P, and K to plants grown from seedlings. Moreover, high frass rates (5 and 10 g dm^−3^) increased the abundance of microbiota responsible for organic matter decomposition. Unlike frass, urea increased the counts of nitrogen-fixing bacteria in peat. The present findings can be used in comprehensive plant growth experiments. Frass is increasingly available due to the growing popularity of insect farms, and it could constitute a promising and environmentally friendly alternative to conventional fertilizers in the near future.

## Figures and Tables

**Figure 1 ijerph-20-00021-f001:**
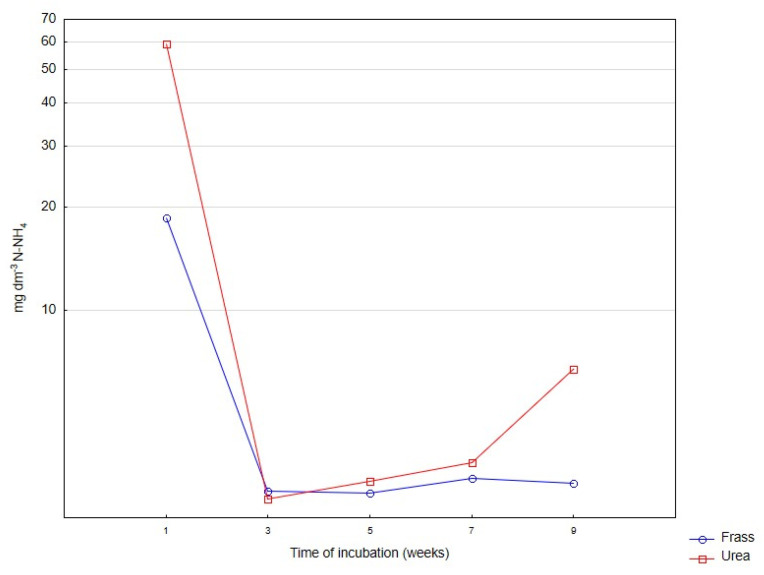
Changes in the content of N-NH_4_ in deacidified peat incubated with frass and urea. Differences are presented on a logarithmic scale.

**Figure 2 ijerph-20-00021-f002:**
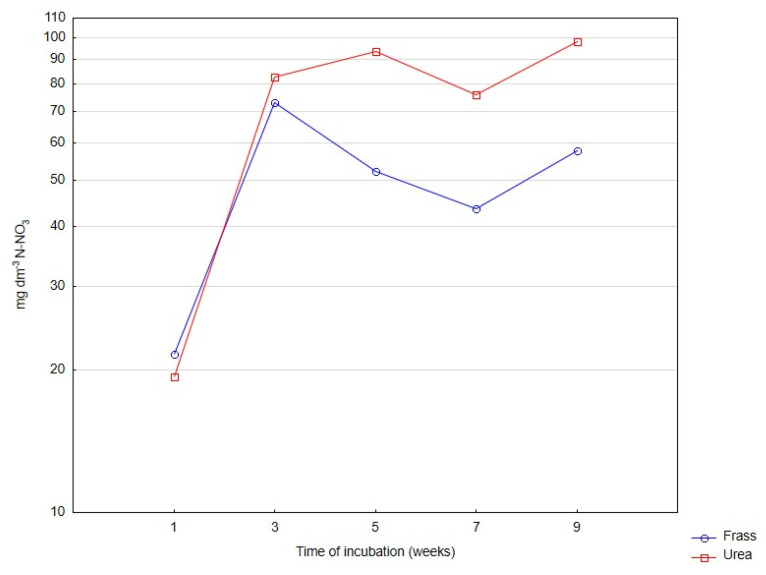
Changes in the content of N-NO_3_ in deacidified peat incubated with frass and urea. Differences are presented on a logarithmic scale.

**Figure 3 ijerph-20-00021-f003:**
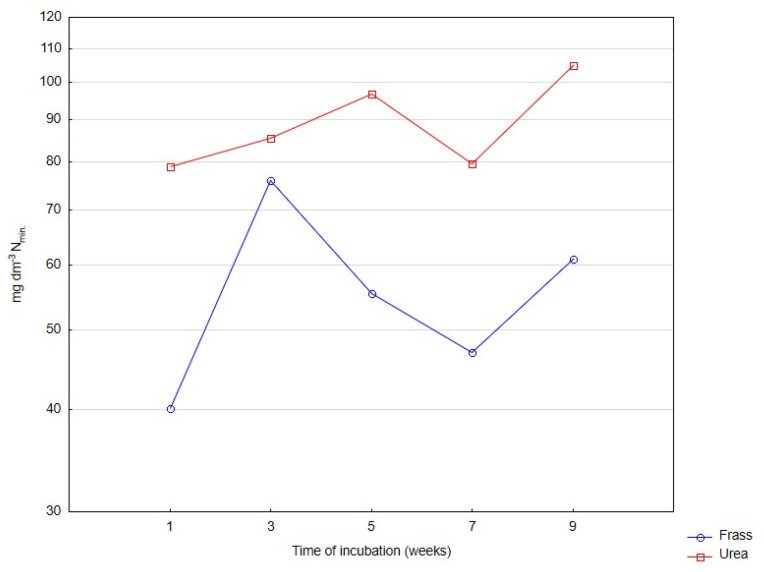
Changes in the content of mineral nitrogen in deacidified peat incubated with frass and urea. Differences are presented on a logarithmic scale.

**Figure 4 ijerph-20-00021-f004:**
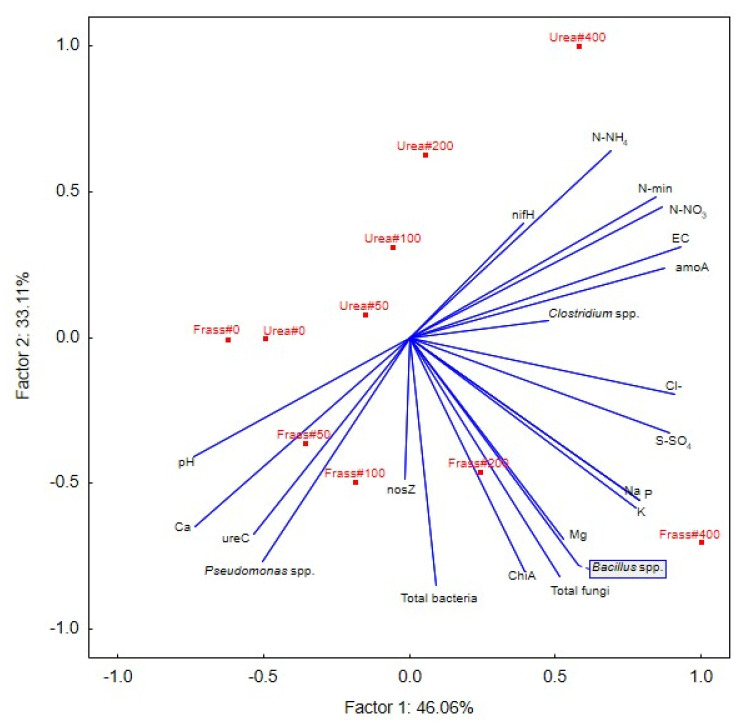
PCA biplot.

**Figure 5 ijerph-20-00021-f005:**
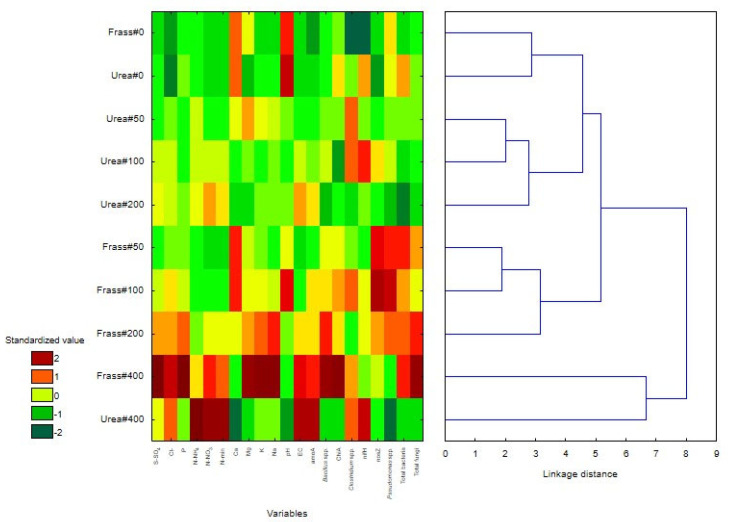
Heatmap with a clustering tree.

**Table 1 ijerph-20-00021-t001:** Average content of mineral nitrogen in deacidified peat supplied with different fertilizers and nitrogen rates.

Fertilizer	Nitrogen Rates (mg dm^−3^)					
	0	50	100	200	400	Mean
N-NH_4_						
Frass	*2.11 a*	*2.36 a*	*3.53 b*	*6.42 c*	*16.45 d*	6.17
Urea	2.40 a	8.00 b	10.35 c	13.54 d	41.69 e	15.19
Significant	ns	*	*	ns	ns	***
Mean	2.25 A	5.18 B	6.94 C	9.98 D	29.07 E	
N-NO_3_						
Frass	*7.90 a*	*16.34 b*	*28.27 c*	*72.96 d*	*122.90 e*	49.7
Urea	3.95 a	26.67 b	58.31 c	92.74 d	188.05 e	73.9
Significant	ns	*	*	*	*	ns
Mean	5.93 A	21.51 B	43.29 C	82.85 D	155.47 E	
N_min_						
Frass	*10.01 a*	*18.70 b*	*31.80 c*	*79.38 d*	*139.34 e*	55.8
Urea	6.35 a	34.67 b	68.66 c	106.28 d	229.73 e	89.1
Significant	ns	*	**	**	****	*
Mean	8.18 A	26.68 B	50.23 C	92.83 D	184.54 E	

Significant differences between nitrogen rates in frass and urea groups: * significant at *p* < 0.05; ** significant at *p* < 0.01; *** significant at *p* < 0.0001; **** significant at *p* < 0.00001; ns—not significant. Capital letters A–E indicate significant differences between nitrogen rates. Lowercase letters a–e indicate significant differences between nitrogen rates only within the frass group (italics) and only within the urea group (without italics).

**Table 2 ijerph-20-00021-t002:** Average content of nutrients and the pH and electrical conductivity (EC) of deacidified peat supplied with different fertilizers and nitrogen rates.

Fertilizer	Nitrogen Rate (mg dm^−3^)					Mean
	0	50	100	200	400	
S-SO_4_						
Frass	20.4	22.8	28.3	35.5	55.7	32.5
Urea	24.1	24.6	28.7	32.2	30.6	28.0
Significant	ns	ns	ns	ns	***	ns
Mean	22.2 A	23.7 A	28.5 AB	33.8 B	43.1 C	
Cl						
Frass	13.2	17.6	20.4	21.2	24.9	19.5
Urea	13.0	17.1	18.6	18.5	21.5	17.7
Significant	ns	ns	ns	ns	ns	ns
Mean	13.1 A	17.4 B	19.5 C	19.9 C	23.2 D	
P						
Frass	19.6	23.7	32.7	50.8	91.1	43.6
Urea	23.2	22.6	21.7	25.5	28.2	24.2
Significant	ns	ns	***	****	****	****
Mean	21.4 A	23.2 B	27.2 C	38.1 D	59.7 E	
Ca						
Frass	1228	1239	1243	1149	1049	1182
Urea	1214	1128	1056	1019	937	1071
Significant	ns	ns	*	ns	*	**
Mean	1221 E	1183 D	1149 C	1084 B	993 A	
Mg						
Frass	80.5	78.5	80.5	84.4	92.8	83.3
Urea	72.6	83.5	77.5	74.4	74.9	76.6
Significant	ns	ns	ns	*	**	**
Mean	76.5 A	81. 0 B	79.0 B	79.4 B	83.8 C	
K						
Frass	43.8	58.5	76.2	107.1	170.6	91.2
Urea	46.7	80.3	52.5	57.5	60.5	59.5
Significant	ns	ns	*	***	****	****
Mean	45.3 A	69.4 C	64.4 B	82.3 D	115.6 E	
Na						
Frass	17.5	19.0	21.7	27.5	33.3	23.8
Urea	18.9	21.4	20.0	20.4	20.0	20.1
Significant	ns	ns	ns	*	**	*
Mean	18.2 A	20.2 B	20.8 B	23.9 C	26.6 D	
pH						
Frass	5.68	5.60	5.70	5.54	5.52	5.61
Urea	5.72	5.54	5.52	5.54	5.46	5.56
Significant	ns	*	**	ns	*	*
Mean	5.7 D	5.57 B	5.61 C	5.54 B	5.49 A	
EC						
Frass	0.10	0.15	0.23	0.54	0.95	0.40
Urea	0.11	0.22	0.38	0.65	1.07	0.49
Significant	ns	*	****	**	*	*
Mean	0.11 A	0.19 B	0.31 C	0.59 D	1.01 E	

Significant differences between frass and urea groups across nitrogen rates: * significant at *p* < 0.05; ** significant at *p* < 0.01; *** significant at *p* < 0.0001; **** significant at *p* < 0.00001; ns—not significant. Capital letters A–E indicate the significance of differences between nitrogen rates.

**Table 3 ijerph-20-00021-t003:** Log-mean values of the microbiological properties of deacidified peat supplied with different fertilizers and nitrogen rates.

Fertilizer	Nitrogen Rate (mg dm^−3^)					Mean
	0	50	100	200	400	
Total bacteria						
Frass	*8.61 b*	*8.71 a*	*8.69 a*	*8.70 a*	*8.71 a*	*8.68*
Urea	8.69 a	8.63 ab	8.61 ab	8.57 b	8.61 ab	8.62
Significant	***	*	*	**	**	***
Mean	8.65 NS	8.67 NS	8.65 NS	8.64 NS	8.66 NS	
Total fungi						
Frass	*11.44 d*	*11.64 b*	*11.56 c*	*11.69 b*	*11.86 a*	*11.64*
Urea	11.48 a	11.48 a	11.46 a	11.43 ab	11.40 b	11.45
Significant	ns	**	ns	**	****	****
Mean	11.46 D	11.56 B	11.51 C	11.56 B	11.63 A	
*Bacillus* spp.						
Frass	*6.91 d*	*6.96 cd*	*7.00 c*	*7.05 b*	*7.14 a*	*7.01*
Urea	6.90 a	6.92 ab	6.96 b	6.88 c	6.89 c	6.91
significant	ns	ns	ns	*	*	**
Mean	6.91 B	6.94 B	6.98 AB	6.96 AB	7.02 A	
*Clostridium* spp.						
Frass	*7.87 b*	*8.03 ab*	*8.14 a*	*8.00 ab*	*8.12 a*	*8.03*
Urea	8.03 b	8.15 a	5.14 a	7.98 b	8.14 a	8.09
Significant	ns	ns	ns	ns	ns	ns
Mean	7.95 C	8.09 B	8.14 A	7.99 BC	8.13 A	
*Pseudomonas* spp.						
Frass	*6.63 c*	*6.87 b*	*7.04 a*	*6.82 b*	*6.37 d*	*6.74*
Urea	6.57 a	6.45 a	6.47 a	6.19 b	5.95 c	6.33
Significant	ns	*	**	**	**	****
Mean	6.60 C	6.66 B	6.75 A	6.51 D	6.16 E	
*chiA*						
Frass	*4.31 c*	*4.37 bc*	*4.42 b*	*4.40 b*	*4.63 a*	*4.43*
Urea	4.41 a	4.32 b	4.21 c	4.27 c	4.24 c	4.29
Significant	ns	ns	*	ns	***	**
Mean	4.36 NS	4.35 NS	4.31 NS	4.34 NS	4.44 NS	
*nifH*						
Frass	*7.46 c*	*7.85 b*	*8.00 a*	*8.06 a*	*7.92 ab*	*7.86*
Urea	8.18 c	7.91 d	8.27 b	7.88 d	8.40 a	8.13
Significant	ns	ns	ns	ns	ns	ns
Mean	7.82 B	7.88 B	8.14 A	7.97 AB	8.16 A	
*amoA*						
Frass	*5.32 d*	*5.41 c*	*5.56 b*	*5.59 b*	*5.67 a*	*5.51*
Urea	5.39 d	5.42 c	5.47 c	5.56 b	5.80 a	5.53
Significant	ns	ns	ns	ns	ns	ns
Mean	5.36 C	5.41 C	5.51 B	5.57 B	5.73 A	
*nosZ*						
Frass	*4.75*	*4.82*	*4.83*	*4.79*	*4.77*	*4.79*
Urea	4.73	4.75	4.79	4.75	4.76	4.76
Significant	ns	**	ns	ns	ns	**
Mean	4.74 B	4.78 AB	4.81 A	4.77 AB	4.76 AB	
*ureC*						
Frass	*6.80 ab*	*6.84 a*	*6.85 a*	*6.82 a*	*6.74 b*	*6.81*
Urea	6.80 a	6.78 a	6.78 a	6.68 b	6.71 b	6.75
Significant	ns	ns	*	**	ns	**
Mean	6.80 A	6.81 A	6.82 A	6.75 B	6.73 B	

Significant differences between nitrogen rates in frass and urea groups: * significant at *p* < 0.05; ** significant at *p* < 0.01; *** significant at *p* < 0.0001; **** significant at *p* < 0.00001; ns—not significant. Capital letters A–E indicate significant differences between nitrogen rates. Lowercase letters a–e indicate significant differences between nitrogen rates only within the frass group (italics) and only within the urea group (without italics).

## Data Availability

Not applicable.

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
