# Peer review of "The Effect of Mealworm Frass on the Chemical and Microbiological Properties of Horticultural Peat in an Incubation Experiment"

_ijerph, 2022, doi:10.3390/ijerph20010021_

Round 1

Reviewer 1 Report

This article presented the effect of mealworm frass on the chemical and microbiological properties of horticultural peat, which is meaningful for future use of insect farming. The study demonstrated that frass can be used as organic fertilizer. Both frass and urea increased the ammonification rate and the nitrification rate, and the abundance of microorganisms. 

This is a research that can support tranfer of insect byproducts into organic use, for which it is considered deserve published. But I found some format problems. Table 1 and 2 can be put into supplementary tables. Table 3 displayed format inconformity with other tables. The tables look too large. Figure 1,2,3 canbe combined into one figure. 

Author Response

Authors’ Response

 We would like to thank for very valuable comments. We improved our manuscript strictly according to all Reviewer’s suggestions. The changes were highlighted in red colour.

Reviewer Comments:

REVIEWER 1

This is a research that can support tranfer of insect byproducts into organic use, for which it is considered deserve published. But I found some format problems.

  1. Table 1 and 2 can be put into supplementary tables.

         Response: Add Table 1 and Table 2 to Supplementary materials.

  1. Table 3 displayed format inconformity with other tables.

         Response: Corrected Table 3 and other.

  1. The tables look too large. Figure 1,2,3 canbe combined into one figure. 

        Response: We would not like to combime because ther will be not readable.

Reviewer 2 Report

I have reviewed this manuscript, and I think it is of important scientific significance, and this manuscript is well written. So I suggest it can be accepted by this journal driectly.

Additional comments to Authors:

1. In the Abstract section, the authors could appropriately provide a comparison of the important study contents between various experiment treatments.

2. If at all possible, the author could further enhance the manuscript's Figures to increase their clarity and make them more aesthetically beautiful.

3. Table 3 has room for improvement and should be consistent with other Tables in this manuscript.

Author Response

Authors’ Response

 We would like to thank for very valuable comments. We improved our manuscript strictly according to all Reviewer’s suggestions. The changes were highlighted in red colour.

Reviewer Comments:

REVIEWER 2

I have reviewed this manuscript, and I think it is of important scientific significance, and this manuscript is well written. So I suggest it can be accepted by this journal driectly.

Additional comments to Authors:

  1. In the Abstract section, the authors could appropriately provide a comparison of the important study contents between various experiment treatments.

Response: Sorry, but we don't really know what the reviewer meant. In our opinion, all comparisons contained in the applications were included in the shortened version in the abstract. If we need to describe something in more detail, we ask the reviewer to give more specific instructions.

  1. If at all possible, the author could further enhance the manuscript's Figures to increase their clarity and make them more aesthetically beautiful.

Response: We chose the most readable version of the drawings, we are limited by the available software.

  1. Table 3 has room for improvement and should be consistent with other Tables in this manuscript.

Response: Corrected Tables in manuscript.
